# How Does E-mail-Delivered Cognitive Behavioral Therapy Work for Young Adults (18–28 Years) with Insomnia? Mediators of Changes in Insomnia, Depression, Anxiety, and Stress

**DOI:** 10.3390/ijerph19084423

**Published:** 2022-04-07

**Authors:** Ayaka Ubara, Noriko Tanizawa, Megumi Harata, Sooyeon Suh, Chien-Ming Yang, Xin Li, Isa Okajima

**Affiliations:** 1Graduate School of Psychology, Doshisha University, Kyoto 610-0321, Japan; uba.a.1229@gmail.com; 2JSPS Research Fellowship, Chiyoda-ku, Tokyo 102-0083, Japan; 3Department of Psychiatry, Shiga University of Medical Science, Otsu 520-2192, Japan; 4Department of Innovation Laboratories, NEC Solution Innovators, Ltd., Tokyo 136-8627, Japan; n-tanizawa@asagi.waseda.jp; 5Public Children Support Center, Adachi-ku, Tokyo 121-0816, Japan; speaknow64@moegi.waseda.jp; 6Department of Psychology, Sungshin Women’s University, Seoul 02844, Korea; alysuh@sungshin.ac.kr; 7Department of Psychology, National Chengchi University, Taipei 11605, Taiwan; yangcm@nccu.edu.tw; 8The Research Center for Mind, Brain & Learning, National Chengchi University, Taipei 11605, Taiwan; 9Department of Psychology, The University of Hong Kong, Hong Kong, China; shirley.li@hku.hk; 10The State Key Laboratory of Brain and Cognitive Sciences, The University of Hong Kong, Hong Kong, China; 11Department of Psychological Counseling, Tokyo Kasei University, Tokyo 173-8602, Japan; 12Faculty of Human Sciences, Waseda University, Saitama 359-1192, Japan

**Keywords:** insomnia, depression, anxiety, stress, cognitive behavioral therapy, mediator, college students

## Abstract

This study examined whether cognitive behavioral therapy (CBT) for insomnia (CBT-I) improved insomnia severity, by changing sleep-related mediating factors. It also examined whether an improvement in insomnia led to enhanced mental health. This study was a secondary analysis of a randomized controlled trial of e-mail-delivered CBT-I for young adults with insomnia. The participants were randomized to either CBT-I or self-monitoring. The mental health-related measures were depression, anxiety, and stress. The sleep-related mediating factors were sleep hygiene practices, dysfunctional beliefs, sleep reactivity, and pre-sleep arousal. A total of 41 participants, who completed all the sessions (71% females; mean age 19.71 ± 1.98 years), were included in the analysis. The hierarchical multiple regression analysis showed that 53% of the variance in the improvements in insomnia severity was explained by the treatment group (*β* = −0.53; Δ*R*^2^ = 0.25; *p* < 0.01) and the changes in sleep reactivity (*β* = 0.39; Δ*R*^2^ = 0.28; *p* < 0.05). Moreover, the mediation analysis showed that the reductions in depression and stress were explained by the changes in insomnia severity; however, anxiety symptoms were not reduced. CBT-I for young adults suggested that sleep reactivity is a significant mediator that reduces insomnia severity, and that the alleviation and prevention of depression and stress would occur with the improvement in insomnia.

## 1. Introduction

College students can experience symptoms of depression and anxiety [1,2]. Insomnia is often persistent [3], and is a risk factor for the subsequent onset of depressive and anxiety disorders [4]. Insomnia is prevalent among adolescents [5,6], with a rate of 22–37% in Asian countries [7,8]. Additionally, approximately 40–65% of patients with anxiety or depressive disorders have insomnia symptoms [9]. This suggests that young adults with sleep disturbances, such as insomnia and sleep debt, are at a higher risk of depression and suicide [1,10].

Studies have revealed that cognitive behavioral therapy (CBT) for sleep disturbances is effective for improving sleep quality and depression among college students [11,12]. For example, a form of CBT for insomnia (CBT-I), delivered by e-mail, called the 8-week Refresh program, may be a cost-effective way for college students with poor sleep quality to improve their sleep quality and reduce depressive symptoms [12]. In Japan, the Refresh program has been effective for young adults with insomnia [13]. It was shown that e-mail-delivered CBT-I had a large effect (Hedges’ *g* = 1.50) on insomnia symptoms, and moderate to large effects on the symptoms of depression, anxiety, and stress (*g* = 0.61, 0.97, and 0.93, respectively), compared to self-monitoring performed by a control group.

Some meta-analyses of CBT-I for patients with comorbid insomnia have suggested that improvements in insomnia symptoms lead to reductions in mental and physical symptoms (e.g., depressive symptoms in depressive disorders and pain symptoms in chronic pain disorders) [14,15]. Therefore, improving insomnia symptoms using CBT-I may contribute to reductions in multiple mental health problems.

However, the process leading to improvements in insomnia and mental health (i.e., depression, anxiety, and stress) has not been sufficiently clarified. Some process factors, such as sleep hygiene practices, sleep reactivity, dysfunctional beliefs, and pre-sleep arousal, can contribute to an improvement in insomnia [16,17,18]. Moreover, several studies have examined the mediating effects of these factors on improving insomnia through CBT-I. Persons et al. [19] conducted a meta-analysis of the mediating effects of the process factors on improving insomnia symptoms by performing CBT-I. Dysfunctional beliefs about sleep and pre-sleep arousal significantly mediated improvements in insomnia symptoms. However, most of the studies included in the meta-analysis involved middle-aged and older individuals, instead of younger populations, and they did not examine the multiple associated factors simultaneously. In addition, the results of the current study add to the extant evidence that the amelioration of insomnia may decrease anxiety and depressive symptoms [20].

This study aimed to clarify the effects of changes in sleep-related mediating factors on the improvement in insomnia, and to examine the effects of improving insomnia on the reduction in symptoms of depression, anxiety, and stress, using CBT-I for young adults with insomnia.

## 2. Materials and Methods

### 2.1. Participants

We conducted a secondary analysis of a randomized controlled trial of e-mail-delivered CBT-I for young adults with insomnia [13]. We recruited participants from 10 January 2018 to 20 December 2018, via advertising at a university campus in Japan. Participants were included if they were university students with a total Insomnia Severity Index (ISI) score ≥ 10, which is the clinical cut-off point for the Japanese version of the ISI [21]. The exclusion criteria included the following: ISI score < 10; current or a history of mental disorders (major depression, bipolar disorder, anxiety disorders, schizophrenia spectrum disorder, eating disorders, attention-deficit hyperactivity disorder, or autism spectrum disorder). After exclusions, the remaining 48 participants (67% females; mean age 19.56 years; standard deviation [SD] ± 1.86 years) were randomly assigned to either the Refresh group (*n* = 24) or the self-monitoring group, who recorded their data in sleep diaries (*n* = 24) for 8 weeks. Forty-one participants (71% females; mean age 19.71 years; SD ± 1.98 years), who completed the intervention, were included in the analysis.

### 2.2. Measures

#### 2.2.1. Insomnia Severity

##### Japanese Version of the Insomnia Severity Index

The ISI is a validated, seven-item, self-reported questionnaire that assesses the severity of insomnia. A summed score was calculated (range 0–28), with higher scores indicating more symptoms of insomnia [22]. The Japanese version of the ISI has been confirmed to have high internal consistency and validity [21].

#### 2.2.2. Insomnia-Related Measures 

##### Japanese Version of the Sleep Hygiene Practices Scale 

The Sleep Hygiene Practices Scale (SHPS) is a validated, 30-item, self-reported questionnaire that assesses the practice of daily living activities and sleep habits that may have negative impacts on sleep. The SHPS was constructed with subscales for sleep schedule and timing, arousal-related behaviors, poor eating and drinking habits prior to sleep, and poor sleep environment. A summed score was calculated (range 30–180), with higher scores indicating poorer sleep hygiene [23]. The Japanese version of the SHPS has been confirmed to have high internal consistency, reliability, and validity [24].

##### Japanese Version of the Ford Insomnia Response to Stress Test

The Ford Insomnia Response to Stress Test (FIRST) is a validated, nine-item, self-reported questionnaire that assesses sleep reactivity to stress (i.e., hyperarousal caused by a stressful event). A summed score was calculated (range 9–36), with higher scores indicating greater sleep reactivity [25]. The Japanese version of the FIRST has been confirmed to have high internal consistency and validity [26].

##### Japanese Version of the Dysfunctional Beliefs and Attitudes about Sleep-16

The Dysfunctional Beliefs and Attitudes about Sleep-16 (DBAS-16) scale is a validated, 16-item, self-reported questionnaire that assesses dysfunctional beliefs and attitudes about sleep. A summed score was calculated (range 0–160), with higher scores indicating more dysfunctional beliefs about sleep [27]. The Japanese version of the scale has been confirmed to have high internal consistency and validity [28].

##### Japanese Version of the Pre-Sleep Arousal Scale 

The Pre-Sleep Arousal Scale (PSAS) is a validated, 16-item, self-reported questionnaire that assesses pre-sleep hyperarousal. A summed score was calculated (range 16–80), with higher scores indicating a state of more hyperarousal [29]. The PSAS was constructed using subscales for somatic arousal (PSASs) and cognitive arousal (PSASc). The Japanese version of the scale has been confirmed to have high internal consistency and validity [30].

#### 2.2.3. Mental Health-Related Measures

##### Japanese Version of the Depression Anxiety Stress Scale-21

The Depression Anxiety Stress Scale-21 (DASS-21) is a validated, 21-item, self-reported questionnaire that assesses depression, anxiety, and stress symptoms to determine mental health [31]. A summed score was calculated (range 0–63), with higher scores indicating more severe mental health. The Japanese version of the scale has been confirmed to have high internal consistency and validity [32].

### 2.3. Procedure

The participants were randomly assigned to either an intervention group or a control group, stratified by sex [13]. In the intervention group, the Refresh program was delivered via e-mail every week, and the participants were required to practice the content and record their data in a sleep diary for 8 weeks. The contents were psychoeducation, stimulus control, relaxation training, mindfulness training, sleep restructuring, and cognitive restructuring. In the control group, a sleep diary was delivered via e-mail every week, and the participants were required to record their data in that sleep diary for 8 weeks. The details of the interventions have been described in a previous study [13]. All the participants were assessed using sleep-related measurements before and after the intervention. 

### 2.4. Statistical Analysis

SPSS Statistics (version 25.0; IBM Inc., Tokyo, Japan), Amos Graphics (version 28.0; IBM Inc., Tokyo, Japan), and R statistical software (version 3.6.3; R Project for Statistical Computing, Vienna, Austria) were used for all the statistical analyses. For all the scales, the changes in scores (before and after treatment) were calculated as delta (Δ) values. During this study, we estimated the effect sizes of the changes in each scale score between the groups, by correcting biases for Hedges’ *g*. In general, an absolute g value of ≥0.2 indicated a small effect size, a value of approximately 0.5 indicated a moderate effect size, and a value of ≥0.8 indicated a large effect size [33].

#### 2.4.1. Descriptive Statistics and *t*-Test

To clarify the effect of the intervention on reducing the symptoms of insomnia and mental health, we conducted a *t*-test of the differences in scores for each scale between the groups.

#### 2.4.2. Hierarchical Multiple Regression Analysis

We performed a hierarchical multiple regression analysis to examine whether the improvements in insomnia severity (ΔISI), resulting from CBT-I, would mediate the effect of the intervention on the changes in insomnia-related symptoms (ΔSHPS subscales, ΔDBAS, ΔPSAS subscales, and ΔFIRST). The intervention group (Refresh or self-monitoring group) was used as the independent variable, and ΔISI was used as the dependent variable during step 1; the independent variables of the ΔSHPS subscales, ΔDBAS, ΔPSAS subscales, and ΔFIRST were added during step 2. An effect size (*R*^2^) of ≤0.02 was considered small, whereas 0.13 was considered moderate, and ≥0.26 was considered large [33].

#### 2.4.3. Mediation Analysis

We performed a series of bias-corrected bootstrap mediation analyses (1000 bootstrap resamples) to examine a model in which the change in insomnia severity (ΔISI) mediates between the intervention and the changes in symptoms of mental health (ΔDASS-depression, ΔDASS-anxiety, and ΔDASS-stress), when a correlation is found between the independent and dependent variables. We used the intervention group as an explanatory variable, the changes in each of the DASS subscales as an objective variable, and the change in insomnia severity as a mediator variable. The hypothetical model of the study is presented in Figure 1. The standardized partial regression coefficient was significant if the 95% bootstrapped confidence interval (CI) did not exceed zero. 

### 2.5. Ethical Considerations

We performed a secondary analysis of a randomized controlled trial of e-mail-delivered CBT-I for young adults [13]. The Refresh study conducted in Japan was approved by the Waseda University Ethics Review Committee (no. 2017-191).

## 3. Results

### 3.1. Comparing Descriptive Statistics between Groups

Table 1 presents the descriptive statistics and results of the *t*-test for each group. The ΔISI (*t* (39) = 3.63; *p* < 0.001) and ΔDASS-anxiety (*t* (39) = 2.97; *p* < 0.05) were significantly lower in the Refresh group than in the self-monitoring group. The ΔDASS-depression (*t* (39) = 1.91; *p* = 0.06) and ΔDASS-stress (*t* (39) = 1.69; *p* = 0.98) were not significantly different between the groups.

### 3.2. Mediators of Improvement in Insomnia Severity 

Table 2 presents the results of the hierarchical multiple regression analysis. The results revealed that 25% of the variance in ΔISI was explained by the treatment (*β* = −0.53; *p* < 0.01) and 28% was explained by ΔFIRST (*β* = 0.39; Δ*R*^2^ = 0.28; *p* < 0.05). The effect size was large (*R*^2^ = 0.53; *F* (9, 31) = 3.86; *p* < 0.05).

### 3.3. Improvement in Insomnia Mediated by Reductions in Depression, Anxiety, and Stress

We performed a correlation analysis between the independent and dependent variables. As a result, there was a significant correlation between the intervention group and ΔDASS-anxiety (*r* = 0.35, *p* < 0.05); ΔDASS-depression and ΔDASS-stress showed a marginally significant correlation (*r* = 0.29, *p* = 0.06; *r* = 0.26, *p* = 0.09, respectively).

The results of the mediation analyses are presented in Figure 2, Figure 3 and Figure 4. Regarding depression symptoms, the mediation analysis showed that the intervention group did not have an effect on ΔDASS-depression (*β* = −0.01; 95% CI: −0.33 to 0.26; *p* = 0.85), but it significantly affected ΔDASS-depression via ΔISI (group to ΔISI: *β* = −0.50, 95% CI: −0.67 to −0.28; *p* < 0.01; ΔISI to ΔDASS-depression: *β* = 0.57, 95% CI: 0.29 to 0.80; *p* < 0.01).

Regarding anxiety symptoms, the mediation analysis showed that the intervention group did not have an effect on ΔDASS-anxiety (*β* = −0.24; 95% CI: −0.65 to 0.15; *p* = 0.21), and that it did not significantly affect ΔDASS-anxiety via ΔISI (group to ΔISI: *β* = −0.50, 95% CI: −0.67 to −0.28; *p* < 0.01; ΔISI to ΔDASS-anxiety: *β* = 0.22, 95% CI: −0.17 to 0.62; *p* = 0.28).

Regarding stress symptoms, the mediation analysis showed that the intervention group did not have an effect on ΔDASS-stress (*β* = 0.01; 95% CI: −0.33 to 0.33; *p* = 0.98), but it significantly affected ΔDASS-stress via ΔISI (group to ΔISI: *β* = −0.50, 95% CI: −0.67 to −0.28; *p* < 0.01; ΔISI to ΔDASS-depression: *β* = 0.54, 95% CI: 0.29 to 0.82; *p* < 0.01).

## 4. Discussion

This study aimed to clarify the effects of changes in sleep-related mediating factors on improving insomnia, and to examine the effects of improving insomnia on the reduction in symptoms of depression, anxiety, and stress, using e-mail-delivered CBT-I for young adults with insomnia. The results of this study revealed that changes in sleep reactivity to stress affected the improvement in insomnia, and that the improvement in insomnia led to a reduction in symptoms of depression and stress, but not symptoms of anxiety.

### 4.1. Mediators of Improvement in Insomnia Severity

Only the changes in sleep reactivity contributed to improving insomnia for young adults with insomnia, who were administered CBT-I. Sleep reactivity is a vulnerable factor in the onset of insomnia, and high sleep reactivity predicts the future onset of depression [25]. Additionally, a previous study reported that changes in sleep reactivity are likely to contribute to the improvement in insomnia severity for patients with insomnia [18]. Therefore, the findings suggest that the effect of CBT-I on the insomnia severity of young adults with insomnia might be mediated by changes in sleep reactivity.

Changes in sleep hygiene behaviors, dysfunctional beliefs, and pre-sleep arousal did not predict changes in insomnia severity. However, previous meta-analyses have concluded that dysfunctional beliefs about sleep and pre-sleep arousal are mediators of insomnia [19]. This occurred because the participants in this study were young adults with insomnia, and because previous studies included participants with ages ranging from 37.1 years (SD ± 12.8) to 70.73 years (SD ± 9.2) in their meta-analyses, and did not examine the influence of sleep reactivity [19]. It is possible that the mechanisms of CBT-I effects may differ between younger individuals and middle-aged or older adults. Therefore, it is necessary to compare the differences in the mediators of CBT-I among generations in the future. 

Okajima et al. [18] simultaneously examined the mediating effects of dysfunctional beliefs and sleep reactivity, as mediators of CBT-I, for middle-aged individuals with insomnia. As a result, only the change in sleep reactivity mediated the improvement in insomnia severity. Because the mediation effect depends on the controlling variables, it is necessary to examine the mediators of insomnia severity, including sleep reactivity, in the future.

### 4.2. Improvement in Insomnia Mediated by Reductions in Depression, Anxiety, and Stress

It was revealed that the changes in insomnia severity improved the symptoms of depression and stress, but not the symptoms of anxiety. CBT-I has been shown to have moderate to large effects on improving psychiatric and somatic symptoms during a meta-analysis of CBT-I for patients with insomnia [14,15]. Similar findings were observed for young adults with insomnia during this study. Bei et al. [34] reported that the increased sleep efficacy contributed to the change in depression among patients with insomnia. Regarding the symptoms of stress, because humans can suppress the secretion of stress hormones during sleep [35], the improvement in stress might be affected by increased sleeping, attributable to the improvement in insomnia. Considering that the reduction in sleep reactivity affects the improvement in insomnia severity, these results suggest that high sleep reactivity contributes to the onset of depression [25]. This suggests that sleep reactivity has a critical and negative role in the severity of insomnia and mental health among young adults.

Although anxiety symptoms decreased in the intervention group, compared with the control group (Hedges’ *g* = −0.71), the improvement in insomnia did not affect this change. We hypothesized that changes in insomnia would affect anxiety; however, the opposite has been shown by a previous study [36] that used a hierarchical multiple regression analysis to examine the effect of improving anxiety symptoms on insomnia severity, using CBT for anxiety. The results of that study indicated that changes in anxious arousal significantly predicted the improvement in insomnia severity, even after controlling for baseline insomnia severity scores; however, the effect of the intervention on insomnia severity was small (Cohen’s d = 0.30) [36]. Therefore, it is possible that other factors, and not the intervention, may have contributed to the decrease in anxiety.

Short et al. [37] proposed that anxiety sensitivity is a transdiagnostic risk factor associated with insomnia symptoms. They examined the effectiveness of interventions for anxiety sensitivity on insomnia severity. Their findings showed that a reduction in anxiety sensitivity was a significant mediator between intervention and insomnia severity. Therefore, it is necessary to investigate anxiety sensitivity as a mediator in the improvement of anxiety symptoms with CBT-I in the future. 

This study had some limitations. First, the sample size was small; however, the sample size was based on a power analysis reported by a previous study [13]. Therefore, further studies, with larger sample sizes, are required to confirm the present results, particularly the mediating effects of insomnia symptoms on improving depression and stress. Second, although this study suggested that changes in sleep reactivity affect the improvement in insomnia, it did not identify which intervention strategies were effective. Therefore, it is necessary to dismantle the intervention studies to clarify the correspondence between the intervention strategies and the changes in the process indicators. Third, only insomnia was measured as a sleep disorder during this study. Although sleep disorders other than insomnia (e.g., sleep irregularity or delayed sleep phase) are more likely to occur among college students [38], they were not measured during this study. Improvements in other sleep disorders may have affected mental health problems. Finally, the validity of the Japanese version of the scales used in this study for younger adults needs to be further examined.

## 5. Conclusions

By conducting e-mail-delivered CBT-I, the changes in sleep reactivity contributed to the improvements in insomnia and, subsequently, depressive symptoms and distress experienced by young adults with insomnia. Despite some limitations, this study is useful for examining the mechanism by which CBT-I improves insomnia and prevents various mental health problems among young adults.

## Figures and Tables

**Figure 1 ijerph-19-04423-f001:**
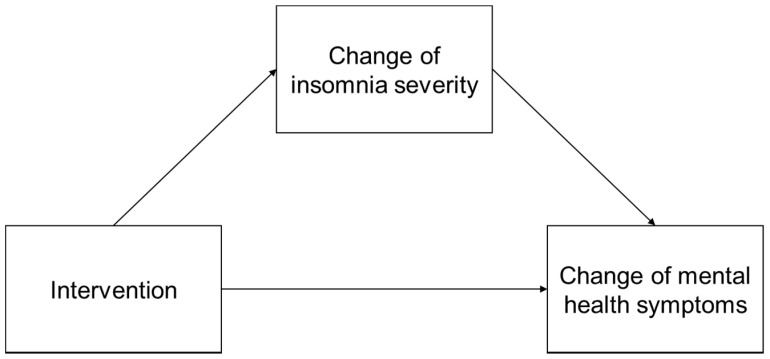
Hypothetical model.

**Figure 2 ijerph-19-04423-f002:**
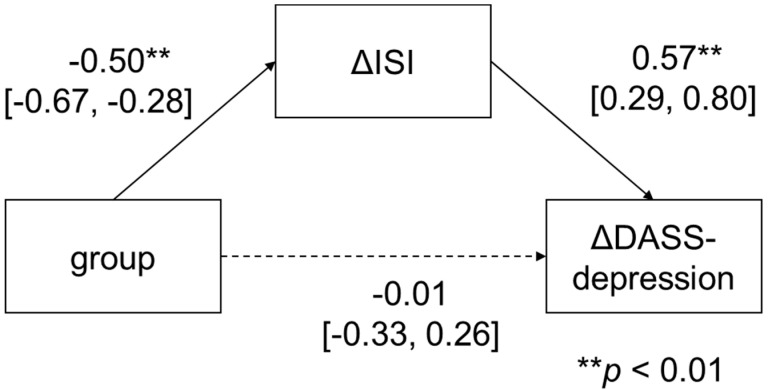
Mediation analysis of DASS-depression.

**Figure 3 ijerph-19-04423-f003:**
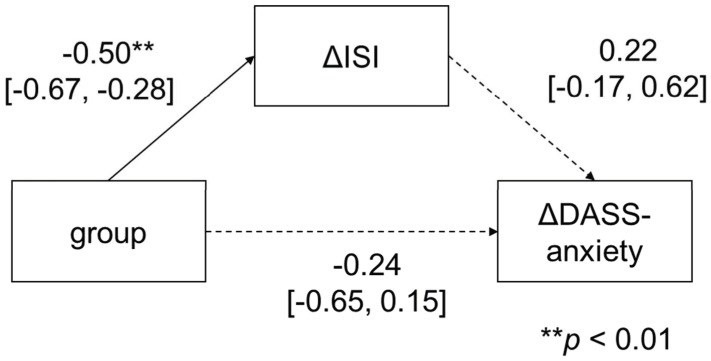
Mediation analysis of DASS-anxiety.

**Figure 4 ijerph-19-04423-f004:**
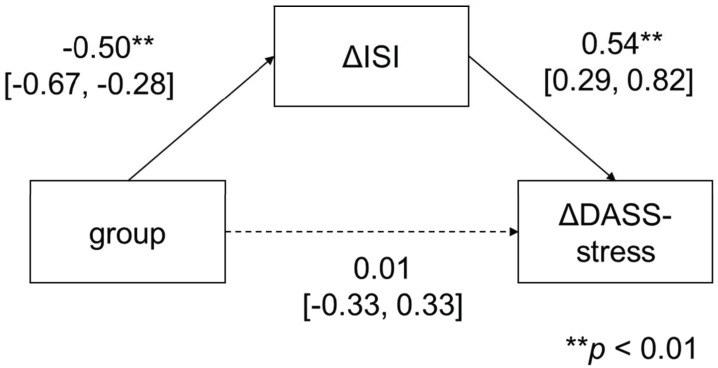
Mediation analysis of DASS-stress.

**Table 1 ijerph-19-04423-t001:** Descriptive statistics of the changes in scores after treatment.

Variables	Intervention Group (*n* = 21)		Control Group (*n* = 20)				Hedges’ *g*
	*M*	*SD*	*M*	*SD*	*t* Value		[95% CI]
**Mental health**							
ΔDASS	−9.52	9.25	−2.90	9.43	2.27	*	−0.70 [−1.31, −0.08]
ΔDASS-depression	−3.19	3.86	−0.75	4.31	1.91		−0.59 [−1.20, 0.03]
ΔDASS-anxiety	−2.38	2.54	−0.50	2.69	2.31	*	−0.71 [−1.32, −0.09]
ΔDASS-stress	−3.67	4.18	−1.55	3.80	1.69		−0.52 [−1.13, 0.09]
**Insomnia symptoms**							
ΔISI	−6.57	4.14	−2.00	3.92	3.63	**	−1.11 [−1.76, −0.46]
**Sleep-related cognitive behavioral factor**							
ΔSHPS total	−17.86	26.38	−2.8	17.92	2.13	*	−0.65 [−1.27, −0.03]
ΔSleep schedule and timing	−7.24	7.64	−1.2	4.44	3.11	**	−0.94 [−1.58, −0.31]
ΔArousal-related behaviors	−6.48	10.06	−1.1	6.87	2.00		−0.67 [−1.29, −0.05]
ΔEating/drinking habits prior to sleep	−3.10	4.64	0.2	4.89	2.21	*	−0.68 [−1.30, −0.06]
ΔSleep environment	−1.05	8.22	−0.7	6.03	0.15		−0.05 [−0.65, 0.55]
ΔDBAS	−17.76	18.96	−7.2	16.47	1.90		−0.58 [−1.20, 0.03]
ΔPSAS total	−11.81	13.47	−2.75	6.97	2.68	*	−0.82 [−1.45, −0.20]
ΔPSASs	−4.10	4.74	0.05	4.44	2.89	*	−0.89 [−1.52, −0.26]
ΔPSASc	−7.71	9.39	−2.8	4.53	2.15	*	−0.65 [−1.26, −0.03]
ΔFIRST	−2.71	5.26	−2.75	3.88	−0.03		0.01 [−0.59, 0.61]

Note: CI = confidence interval, DASS = Depression Anxiety Stress Scale, DASS-depression = Depression Anxiety Stress Scale for depression, DASS-anxiety = Depression Anxiety Stress Scale for anxiety, DASS-stress = Depression Anxiety Stress Scale for stress, ISI = Insomnia Severity Index, M = mean, SD = standard deviation, SHPS = Sleep Hygiene Practices Scale, DBAS = Dysfunctional Beliefs and Attitudes about Sleep Scale, PSAS = Pre-Sleep Arousal Scale, PSASs = Pre-Sleep Arousal Scale—somatic, PSASc = Pre-Sleep Arousal Scale—cognitive, FIRST = Ford Insomnia Response to Stress Test. * *p* < 0.05. ** *p* < 0.01.

**Table 2 ijerph-19-04423-t002:** Results of the multiple regression analysis.

Variables		B	SE	*β*		*R* ^2^		Δ*R*^2^	
ΔISI									
Step1:	group	−2.00	0.90	−0.50	**	0.25	**	0.25	**
Step2:	group	−4.79	1.61	−0.53	**	0.53	*	0.28	*
	ΔSleep schedule and timing	0.17	0.13	0.26					
	ΔArousal-related behaviors	−0.10	0.11	−0.19					
	ΔEating/drinking habits prior to sleep	0.13	0.18	0.14					
	ΔSleep environment	0.16	0.13	0.25					
	ΔDBAS	−0.06	0.04	−0.23					
	ΔPSASs	−0.13	0.16	−0.14					
	ΔPSASc	−0.01	0.12	−0.01					
	ΔFIRST	0.39	0.18	0.39	*				

Note: B = partial regression coefficient, *β* = standardized partial regression coefficient, DBAS = Dysfunctional Beliefs and Attitudes about Sleep Scale, FIRST = Ford Insomnia Response to Stress Test, ISI = Insomnia Severity Index, PSASc = Pre-Sleep Arousal Scale—cognitive, PSASs = Pre-Sleep Arousal Scale—somatic, *R* = coefficient of determination, *R*^2^ = *R*-squared, SE = standard error. * *p* < 0.05. ** *p* < 0.01.

## Data Availability

The datasets analyzed in the current study are available from the corresponding author, upon reasonable request.

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
