# Peer review of "How Does E-mail-Delivered Cognitive Behavioral Therapy Work for Young Adults (18–28 Years) with Insomnia? Mediators of Changes in Insomnia, Depression, Anxiety, and Stress"

_ijerph, 2022, doi:10.3390/ijerph19084423_

Round 1

Reviewer 1 Report

I am reviewing “How Does e-mail-Delivered Cognitive Behavioral Therapy Work for Young Adults with Insomnia? Mediators of Changes in Insomnia, Depression, Anxiety, and Stress” for the International Journal of Environmental Research and Public Health.  I loved what the authors were doing, but they have a fatal flaw in their study.  Mediation demands that the predictor is related to the criterion/criteria, the predictor is related to the mediator(s), and the mediator(s) is related to the criterion/criteria.  The authors did not show that the predictor was related to the criteria in the form of change in depression and change in stress, so mediation cannot be demonstrated.  However, mean differences were found in these criteria variables across the two groups, so it is likely that the error variance was large due to the small sample.  I recommend that the authors triple or quadruple their sample or run a power analysis to determine the proper sample needed and rerun their analyses.  With such a sample, I think the authors would find the mediating relations that they say they find in the present article and I would be happy to read and approve such an article.

Author Response

Response:

Thank you for your suggestion. We agree with your comment about the small sample size. In order to address the limitation of the small sample size, we used a series of bias-corrected bootstrap mediation analyses (1000 bootstrap resamples). Bootstrapping is a computationally intensive method that involves repeated sampling from the data set in each resampled data set (Preacher & Hayes, 2008). Bootstrapping also mitigates sample error for small sample sizes (MacKinnon, Lockwood, Williams, 2004). Therefore, we believe that the use of the bootstrap method solved the problem you pointed out.

Changes:

Lines 178-186

2.4.3. Mediation analysis

 We performed a series of bias-corrected bootstrap mediation analyses (1000 bootstrap resamples) to examine a model in which the change in insomnia severity (ΔISI) medi-ates between the intervention and the changes in symptoms of mental health (ΔDASS-depression, ΔDASS-anxiety, and ΔDASS-stress). We used the intervention group as an explanatory variable, the changes in each of the DASS subscales as an ob-jective variable, and the change in insomnia severity as a mediator variable. The hypothetical model of the study is presented in Figure 1. The standardized partial regression coefficient was significant if the 95% bootstrapped confidence interval (CI) did not ex-ceed zero.

Lines 211-225

3.3. Improvement in insomnia mediated by reductions in depression, anxiety, and stress

  The results of the mediation analyses are presented in Figures 2–4. Regarding depression symptoms, the mediation analysis showed that the intervention group did not have an effect on ΔDASS-depression (β = -0.01; 95% CI: -0.33 to 0.26; p = 0.85), but it significantly affected ΔDASS-depression via ΔISI (group to ΔISI: β = -0.50, 95% CI: -0.67 to -0.28; p < 0.01; ΔISI to ΔDASS-depression: β = 0.57, 95% CI: 0.29 to 0.80; p < 0.01).

Regarding anxiety symptoms, the mediation analysis showed that the intervention group did not have an effect on ΔDASS-anxiety (β = -0.24; 95% CI: -0.65 to 0.15; p = .21), and that it did not significantly affect ΔDASS-anxiety via ΔISI (group to ΔISI: β = -0.50, 95% CI: -0.67 to -0.28; p < 0.01; ΔISI to ΔDASS-anxiety: β = 0.22, 95% CI: -0.17 to 0.62; p = 0.28).

  Regarding stress symptoms, the mediation analysis showed that the intervention group did not have an effect on ΔDASS-stress (β = 0.01; 95% CI: -0.33 to 0.33; p = .98), but it significantly affected ΔDASS-stress via ΔISI (group to ΔISI: β = -0.50, 95% CI: -0.67 to -0.28; p < 0.01; ΔISI to ΔDASS-depression: β = 0.54, 95% CI: 0.29 to 0.82; p < 0.01).

Reviewer 2 Report

Dear authors, it was a pleasure to read and revise this manuscript. This work presents a secondary analysis on the effects of an online CBTi program for yongsters. Although it is well presented, a few suggestions can be made.

title

-youngsters: provide age

methods

-line 90 :(such as bipolar or....==> only 2 options?

-line 91: of mental disorders ==> such as...?

-line 106: all questionnaires were translated in to Japanese. I see there are references linked. In what way are these translated questionnaires validated in use in younger adults?

-path analysis: why not use mediation analysis?

-path analysis: please already include figure 1 here already (without results), that way you can nam/label the arrows which helps the reader to understand what analysis you want to conduct.

-table 1: please provide in legend==> M=mean?; SD=Standard deviation? ?....

-table 1: please make sure the 95% CI stands on one line, now there is not enough room in this column which makes it harder to read (lay-out)

-table 2: please add in legend: B=...?; SE=...?beta=...., R and R²=.....

-figure 1: please split figure 1 in to two, part one is single regression and part two is multiple regression with extra arrow (mediation).

-what is the rationale in this analysis to conduct a mediation analysis if the single regression did not turn out to be significant? (I.E in DASS depression and in DASS stress)

-lines 269....: were any covariates included in the model? If so, which ones, if not, motivate why certain covariates were not included (i.e smoking, screen time, medication, cafeine,smoking,physical activity, sedentary behaviour...)

-please provide more detail on the online CBTi program itself in the methods, or provide it in an attachment and figure

Round 2

Reviewer 1 Report

The authors used a bootstrapping technique instead of increasing their small sample by 3 or 4 times.  The bootstrapping technique did not establish relations between the independent variable and two of the dependent variables (change in depression and change in stress), which is the fatal flaw of the paper.  The absence of these relations can be found in Table 1.  As the authors should know, one cannot demonstrate mediation of relations that do not exist.  
